# Interactive Visualization Recommendation with Hier-SUCB

## ABSTRACT

Visualization recommendation aims to enable rapid visual analysis of massive datasets. In real-world scenarios, it is essential to quickly gather and comprehend user preferences to cover users from diverse backgrounds, including varying skill levels and analytical tasks. Previous approaches to personalized visualization recommendations are non-interactive and rely on initial user data for new users. As a result, these models cannot effectively explore options or adapt to real-time feedback. To address this limitation, we propose an interactive personalized visualization recommendation (**PVisRec**) system that learns on user feedback from previous interactions. For more interactive and accurate recommendations, we propose **Hier-SUCB**, a contextual combinatorial semi-bandit in the PVisRec setting. Theoretically, we show an improved overall regret bound with the same rank of time but an improved rank of action space. We further demonstrate the effectiveness of **Hier-SUCB** through extensive experiments where it is comparable to offline methods and outperforms other bandit algorithms in the setting of visualization recommendation.

## CCS CONCEPTS

• **Information systems** → **Personalization**; *Retrieval models and ranking*; • **Human-centered computing** → *Visualization systems and tools.*

**ACM Reference Format:**
Anonymous Author(s). 2024. Interactive Visualization Recommendation with Hier-SUCB. In . ACM, New York, NY, USA, 9 pages. https://doi.org/10.1145/nnnnnnn.nnnnnnn

## 1 INTRODUCTION

Data visualization has been widely used across domains (i.e., public health[14], engineering[3], social science[11]) to analyze data and obtain insights for effective communication and decision-making. With a suitable visualization, users can observe the tendency of one variable or the correlation of multiple variables. It is also more intuitive for users to choose from generated visualizations than from raw data [31]. Driven by the intuitive nature of visualizations in data exploration, visualization recommendation systems aim to enhance user experience by accelerating the analysis and exploration of large datasets. In practice, different users usually have different backgrounds and purposes (e.g., skill levels, analytic tasks). Thus, it is highly desired to quickly receive and understand

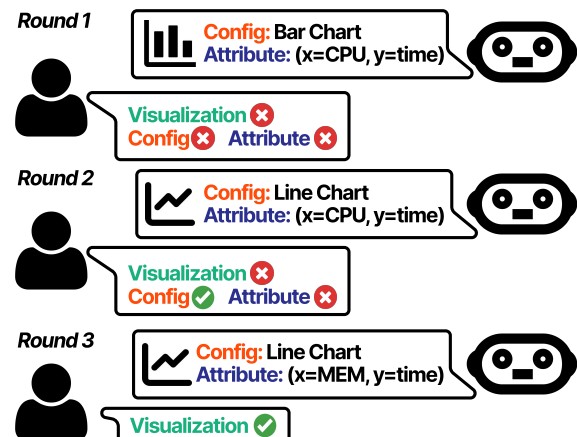

**Figure 1: An illustrative example of interactive PVisRec. In this example, a software engineer wants to quickly find a useful visualization to understand a system log dataset. In Round 1, the agent first recommends a visualization and receives feedback from the user on the entire visualization. If the agent receives negative feedback, it will further ask for the user's preference on the attributes and the configuration separately, since it is possible that the user prefers the attributes and the configuration, but dislikes their visualization (as shown in Round 2). With the received feedback, the agent will update the model and recommend a high-quality and useful visualization for the user in Round 3.**

the user preferences to adapt the recommendation model for more personalized visualization recommendations [21].

Previous works on personalized visualization recommendation [22, 27, 21, 31] are usually non-interactive and heavily rely on the user embeddings for cold-start of new users. For a new user without any background knowledge, previous works [26, 27] recommend visualizations based on the averaged user embedding, and thus not able to capture the actual user's personalized preferences for visualizations. For example, a software engineer wants to understand the resource usage of a program over time, but the system log contains so many variables that finding the best visualization would be overly time-consuming. With traditional methods, the agent can recommend the most significant visualizations based on the statistical features of the variables, but fails to give more accurate recommendations that rely on the user intention revealed during the interactions. As a result, their models can not efficiently explore the search space and quickly adapt to the user's real-time feedback in an interactive fashion.

In this paper, we propose an interactive system for rapid and personalized visualization recommendation. As shown in Fig. 1, by utilizing the user feedback from previous interactions, the agent can efficiently explore and narrow down the search space rapidly to recommend better visualizations. It is non-trivial to modify and apply bandit algorithms to the PVisRec Problem. *Traditional contextual bandit algorithms* [5, 17] usually suffer from a large action

space in the PVisRec setting where users give feedback on a combination of items and visualizations. Compared with combinatorial bandits[28], non-combinatorial bandits will have an action space with higher ranks, since recommending a visualization is equivalent to recommending attributes and one configuration. *Contextual Combinatorial bandit algorithms* [28, 9] provide a reduced action space in the recommendation of visualizations, but they suffer from the gap between the real reward and the estimated reward. In the PVisRec setting, a user may like an attribute pair and a configuration but dislike their combination as a visualization, which creates a novel kind of bias in the estimation of the reward. *Semi-bandit algorithms* [18, 23] introduce bias terms to solve the problem of reward estimation, but their combinatorial implementation relies on cascading assumptions, which conflicts with our setting where a user rates the items in combination independently. Plus, their bias terms are unlearnable, thus these algorithms fail to model the underlying relation between configurations and attributes.

To provide an interactive personalized visualization recommendation system, we introduce **Hier-SUCB**, a contextual combinatorial semi-bandit with a hierarchical structure tailored to the PVisRec problem. To narrow the gap between the real and estimated reward, we model the bias in estimation by proposing *an additional bias term* in the exploration of attributes to represent the relation between attributes and configurations. Different from previous work on semi-bandits, our bias is learnable and estimates the interrelation of the visualization with an independent bandit. To apply the bias term to PVisRec, we construct *a hierarchical bandit structure* that gets user feedback flexibly for visualizations, configurations, and attributes. The agent will decide on a configuration before evaluating the whole visualization that contains the bandit of bias term with larger action space, and thus accelerate the learning of our agent. With the hierarchical design and bias term in our algorithm, we improve the regret bound from $O(\sqrt{Tln^3(nm^2Tln(T))})$ by SPUCB [23] to $O(\sqrt{Tln^3(m^2Tln(T))})$ where $T$ is overall rounds, $n$ is the number of configurations and $m$ is the number of attributes. We test the proposed hierarchical algorithm based on a synthetic experiment and a simulated real-world experiment validated by human evaluation.

To summarize, our contributions are:

- We propose an interactive method that learns from real-time user input, avoiding the need for initial data and enabling personalization even in cold-start cases.
- We propose a learnable bias term to model configuration-attribute relations, with a hierarchical structure to enhance user experience and accelerate learning.
- We theoretically prove the superiority of **Hier-SUCB** with a lower regret bound and confirm it through experiments, outperforming other bandit algorithms and the offline method.

## 2 RELATED WORK

### 2.1 Visualization Recommendation

The first work on visualization recommendation focused entirely on simple rules defined by experts [19, 10]. It was followed by other rule-based systems including Voyager [32, 33, 34], VizDeck [24], DIVE [13], and many other systems [4, 12]. Such rule-based systems leverage a large set of rules for recommending visualizations and do not consider any learning or user personalization. Recently, some work has focused on the ML-based visualization recommendation problem [26, 8, 6, 15]. Qian et al. [27] introduced the problem of personalized visualization recommendation where user-level models are learned for recommending visualizations that are personalized based on past user interactions along with the data and configurations of those visualizations. They solve the problem of sparse user feedback by introducing the notion of meta-features to leverage feedback from visualizations across different datasets. However, these works focused on learning user-agnostic models for visualization recommendation, and thus are unable to be used for the personalized visualization recommendation problem. Other work such as VizRec [21] is only applicable when there is a single dataset shared by all users, and the users have explicitly liked and tagged such visualizations, as the approach also uses meta-data. Song et al. [29] further proposed a hybrid retrieval-generation framework for data visualization systems. Ojo et al. [22] represented the corpus of datasets and the visualizations as a large graph and proposed a graph neural network approach to achieve better personalized visualization recommendations, while Haotian et al. further represent visualization embedding in knowledge graphs [16]. However, these existing solutions are all offline and have a relatively low hit rate when recommending the top 1 visualization to the user. They are also not capable of learning user preferences through real-time interaction.

### 2.2 Contextual Combinatorial Bandit

Combinatorial bandit is a common way to solve the recommendation of multiple items in the online recommendation setting. Qin et al. [28] formulated a combinatorial bandit in the contextual bandit setting, which combines the user feature vector in the contextual bandit and the reward function from combinatorial bandits. In each turn, the user is recommended a combination of items, whose reward is assumed to be the average of the related item reward. Some work by Gin et al. [28] extended the average function to any non-linear function that is monotonic and Lipschitz continuous. Based on the contextual bandit setting, Peng et al. [23] introduced semi-bandit with bias to model dynamic feature vectors over time. Combinatorial recommendation is also discussed in this work, but with a cascading bandit assumption that was originally discussed in work by Li et al. [18]. However, in the PVisRec setting, the reward given by the combination cannot be well represented by a monotonic setting, since the user preference of the visualization not only depends on the data attribute and visual configuration, but also depends on their relation when combined. Also, the cascading assumption is unrealistic in our personalized visualization recommendation setting, since there is no given mapping between the user feedback of a configuration and the attributes. Furthermore, the bias in previous semi-bandit works is neither learnable nor assigned to specific combinations, and therefore, fails to model the interrelation of visualization in PVisRec problem.

### 2.3 Bundle Recommendation

Previous work by Deng et al [7] proposed a policy-based reinforcement learning (RL) approach that utilizes the user embedding and

prior knowledge of items as a graph to recommend a bundle of items to users. The user embedding is carefully studied and selected, and the cold start of new users relies on the similarity of the new user to previous users. Chang et al. [2] modeled the interactions between users and items/bundles as well as the relations between items and bundles by a graph neural network. Noticing the bias caused by submodular functions, Mehrotra and Vishnoi [20] give a theoretical analysis of bias in subset selection in recommendation systems However, the visualization to be recommended to the user in the personalized visualization recommendation setting is often unique, as it is fundamentally tied to the attributes in the users' dataset being visualized along with the visual configuration. This differs from previous works [7, 2] in bundle recommendation that explicitly assumes the items in any such bundle being recommended are shared by all users. Additionally, in personalized visualization recommendation, the user embedding related to the preference is too complex to be explicitly modeled by simple labels (*e.g.*, gender, level of education, and age) in bundle recommendation. Furthermore, in bundle recommendation, the relation between items and bundles differs from the interrelation among items, which is more important in the PVisRec problem.

## 3 PROBLEM FORMULATION

In this section, we formulate interactive personalized visualization recommendation as a contextual combinatorial bandit problem with bias. We start with the traditional contextual combinatorial bandit scenario where an agent recommends a combination of items to a user and receives feedback for the combination. We then discuss the limitations of models mentioned in the related works, and propose a new hierarchical structure and bias term to improve the recommendation in PVisRec setting.

### 3.1 Contextual Combinatorial Bandit

Contextual combinatorial bandit algorithms are applied in various settings. In previous work done by Qin et al. [28], the combination is assumed to have a reward that is a linear combination of rewards from related items, which is incompatible with visualization recommendation. Furthermore, previous works on contextual combinatorial bandits do not consider recommending items from more than one category, which is a critical aspect of the visualization recommendation problem.

To overcome these issues, we formulate a novel contextual bandit problem that allows the recommender agent to choose 3 items from 2 different categories: configuration, attribute of the x-axis and attribute of the y-axis. Different from the former contextual bandit settings [28], the agent has to select exactly one item from each category. We consider the stochastic k-armed contextual bandit problem from the 2 different categories, where the total number of rounds $T$ is known. For each round $t$, user plays action $a_t = (a_{c,t}, a_{x,t}, a_{y,t})$, picking one configuration and two attributes. Picking the configuration is noted as $a_{c,t}$, picking x-axis attribute is noted as $a_{x,t}$ and picking y-axis attribute is noted as $a_{y,t}$.

For the category of configuration, we assume there are $n$ items. We denote the feature vector of configuration in round $t$ as $\mathbf{x}_{c,t}$. The feature vector is generated based on the historical data of visualization recommendation using collaborative filtering.

For the category of attributes, we assume there are $m$ items and we need to select two items (x-axis and y-axis) each round. We denote the feature vector of x-axis and y-axis in round $t$ as $\mathbf{x}_{x,t}, \mathbf{x}_{y,t}$. The feature vector is generated based on the statistical values extracted from the data points of this attribute, as done in [26].

Following the definition of contextual bandits and the denotation above, in round $t$, the reward of configuration $r_{C,t}$, the reward of attribute $r_{A,t}$ and the reward of visualization $r_{V,t}$ is modeled as:

$$r_{C,t} = \langle \theta_{C,t}, \mathbf{x}_{c,t} \rangle + \epsilon_{t,a_t} \tag{1}$$

$$r_{A,t} = \langle \theta_{A,t}, \mathbf{x}_{x,t} \rangle + \langle \theta_{A,t}, \mathbf{x}_{y,t} \rangle + \epsilon_{t,a_t} \tag{2}$$

$$r_{V,t} = r_{C,t} + r_{A,t} \tag{3}$$

where $\epsilon_{t,a_t}$ is a noise term with sub-Gaussian distribution and zero mean. We follow conventional assumption in bandit problems [30] that assume the inconsistent user feedback can be simplified with a sub-Gaussian distribution with zero mean. $\theta_{A,t}$ and $\theta_{C,t}$ are learnable parameters for the estimation of users' preference for attributes and configurations. Due to the hierarchical structure, the feature vectors of configurations have smaller dimensions than the feature vectors of attributes, so we use $\theta_{C,t}$ to separately denote user preference for configurations.

To reduce calculation, we use the same $\theta_{A,t}$ to denote user preference for x-axis and y-axis attributes as they have same feature vector dimensions. Using the same set of parameter will not affect the estimation. According to the contextual bandit setting, we assume the reward of visualization is a linear combination of the configuration reward, x-axis reward, y-axis reward and bias term. If we use different parameters $\theta_{x,t}, \theta_{y,t}$ for x-axis and y-axis, their sum would be $< \theta_{x,t}, \mathbf{x}_{x,t} > + < \theta_{y,t}, \mathbf{x}_{y,t} >$. But we can always concatenate $\theta_{x,t}, \theta_{y,t}$ to be $\theta_{A,t} = \theta_{x,t}^{(1)}, \theta_{x,t}^{(2)}, ..., \theta_{x,t}^{(n)}, \theta_{y,t}^{(1)}, \theta_{y,t}^{(2)}, ..., \theta_{y,t}^{(n)}$, which has double number of dimension compared to the parameter of x-axis or y-axis. Meanwhile, we concatenate zero vector to $\mathbf{x}_{x,t}, \mathbf{x}_{y,t}$ so that the new vector $\mathbf{x}'_{x,t}, \mathbf{x}'_{y,t}$ would be $x_1, x_2, ..., x_n, 0, 0, ..., 0$ and $0, 0, ..., 0, y_1, y_2, ..., y_n$. Calculating the inner value between $\theta_{A,t}$ and $\mathbf{x}'_{x,t}, \mathbf{x}'_{y,t}$ would be the same as using two parameters $< \theta_{x,t}, \mathbf{x}_{x,t} > + < \theta_{y,t}, \mathbf{x}_{y,t} >$.

We further represent the parameters $\theta_{c,t}, \theta_{x,t}, \theta_{y,t}$ with matrix $V_{C,t}, V_{A,t}$ and vector $b_{C,t}, b_{A,t}$. They are initialized with:

$$\theta_{C,t} = V_{C,t} + b_{C,t} = \mathbf{I}_d + \mathbf{0}_d \tag{4}$$

$$\theta_{A,t} = V_{A,t} + b_{A,t} = \mathbf{I}_d + \mathbf{0}_d \tag{5}$$

In each turn $\theta_{C,t}, \theta_{A,t}$ is updated by updating $V_{C,t}, b_{C,t}$ and $V_{A,t}, b_{A,t}$.

$$V_{C,t} = V_{C,t-1} + \mathbf{x}_{c,t-1}\mathbf{x}_{c,t-1}^T \tag{6}$$

$$V_{A,t} = V_{A,t-1} + (\mathbf{x}_{x,t-1} + \mathbf{x}_{y,t-1})(\mathbf{x}_{x,t-1}^T + \mathbf{x}_{y,t-1}^T) \tag{7}$$

$$b_{C,t} = b_{C,t-1} + r_{C,t-1}\mathbf{x}_{c,t-1} \tag{8}$$

$$b_{A,t} = b_{A,t-1} + r_{A,t-1}(\mathbf{x}_{x,t-1} + \mathbf{x}_{y,t-1}) \tag{9}$$

The goal of the agent is to minimize the cumulative regret and maximize the average reward in T rounds through repeated item combination recommendations. With the reward defined above, we want to optimize a cumulative regret defined as follows:

$$Reg(T) = \sum_{t=1}^{T} (r_t^* - r_{t,a_t}) \tag{10}$$

where $r_t^*$ refers to optimal reward in round $t$.

In contextual combinatorial bandit, the reward of target combination is assumed to be a linear combination of reward of its items, which is unrealistic in the PVisRec setting. For the finite-dimensional feature vectors of attributes, it is impossible to find parameter $\theta$ that ensures $\epsilon$ is a zero-mean random variable [23]. More specifically, in the PVisRec problem user may like both the attributes and configuration but not their combination. To allow the agent to be aware of this difference, we need to have a different set of parameter $\theta_A$ for the attribute pair $A_{1,2}$, which is a fundamental contradiction in the contextual bandit setting[28].

## 3.2 Bias Term in PVisRec

We introduce a learnable bias term in the definition of reward function. It is an additional term in the reward function that is learnable and represents the relation between configuration and attributes. Based on Eq. 3 above, we rewrite the reward function of visualization in round $t$ with action $a_t$ as:

$$r_V(t, a_t) = \langle \theta_{C,t}, \mathbf{x}_{c,t} \rangle + \langle \theta_{A,t}, \mathbf{x}_{x,t} \rangle + \langle \theta_{A,t}, \mathbf{x}_{y,t} \rangle + r_{\gamma,t} + \epsilon_{t,a_t} \quad (11)$$

where $\gamma$ is an arm-specific bias term and $\epsilon_{t,a_t}$ sub-Gaussian noise term with zero mean. We model the bias term as a simple multi-armed bandit that has the same action space as the visualization given configuration $C_k$, attribute pair $A_i, A_j$. We assume the reward of bias term $r_{\gamma,t}$ as a function of visualization reward, configuration reward and attribute reward:

$$r_{\gamma,t} = f(r_{V,t}, r_{C,t}, r_{A,t}) \quad (12)$$

There could be various non-linear functions $f(r_V, r_C, r_A)$, and the rewards in this equation follow the definition of Bernoulli reward $r_{C,t}, r_{A,t}, r_{V,t} \in \{0, 1\}$.

## 3.3 Hierarchical Structure in PVisRec

In PVisRec problem, we derive a hierarchical structure both in the interaction with users and the learning process of the agent. The user is first asked to determine whether visualization $V_t$ is favorable by giving a Bernoulli feedback $r_{V,t} \in \{0, 1\}$. If the answer is positive, the reward of configuration is $r_{C,t} = 1$ and the reward of the attribute pair is $r_{A,t} = 1$. However, if the answer is negative, then the user will be further queried for the preference of configuration and attributes to get $r_{C,t}, r_{A,t}$. In this system, the user provides feedback for both the combinatorial contextual bandits and the bias term bandits without degrading the user experience. By evaluating the bias term with the feedback on the entire visualization, we avoid the cascading assumption that would otherwise harm the user experience.

The agent in our algorithm is also implemented with a hierarchical structure. In each round $t$ the agent will first decide the optimal action in configuration $a_{c,t}$, and then decides the $a_{x,t}, a_{y,t}, \gamma_t$ based on the summed upper confidence bound of configuration reward, attribute reward and the bias. Since the configuration usually has a smaller arm pool as studied in Fig. 3, the agent only needs to estimate the reward of attributes and the bias term for the majority of time. With such breakdown the agent is free from $O(mn^2)$ action space in the bias term, and can thus work with lower regrets.

## 4 ALGORITHM & THEORETICAL ANALYSIS

In this section, we propose our combinatorial bandit algorithm for interactive personalized visualization recommendation, called Hierarchical Semi UCB (Hier-SUCB). The code is available at https://github.com/HierSUCB-WWW/HierSUCB/tree/main.

### 4.1 Hier-SUCB

Inspired by SPUCB [23], we develop a combinatorial contextual semi-bandit with a learnable bias term and a hierarchical structure. The structure includes a hierarchical agent to optimize the exploration in biased combinatorial setting and a hierarchical interaction system to get detailed user feedback without hurting user experience. The algorithm maintains two sets of upper confidence bounds (UCB) including the UCB of configurations $U(c)$ and visualizations $U(c, x, y)$ with a given configuration $c$. More formally, let $U(c)$ and $U(v) = U(c, x, y)$ be defined as:

$$U(c_t) = \theta_{C,t}^T \mathbf{x}_{c,t} + \rho_{c,t} \quad (13)$$

$$U(v_t) = \theta_{C,t}^T \mathbf{x}_{c,t} + \theta_{A,t}^T (\mathbf{x}_{x,t} + \mathbf{x}_{y,t}) + \gamma_t + \rho_{c,t} + \rho_{a,t} + \rho_{\gamma,t} \quad (14)$$

The confidence radius of attribute and configuration $\rho_a, \rho_c$ is defined as:

$$\rho_{k,t} = \sqrt{\mathbf{x}_{k,t}^T (\mathbf{I}_d + \mathbf{x}_{k,t} \mathbf{x}_{k,t}^T) \mathbf{x}_{k,t}}, k \in \{a, c\} \quad (15)$$

where $x_t^T$ is the embedding vector of attribute or configuration in round $t$ and $\mathbf{I}_d$ refers to identity matrix with the same dimension $d$ as $\mathbf{x}_t$. According to UCB[1], the confidence radius of bias $\rho_\gamma$ defined as

$$\rho_{\gamma,t} = \sqrt{2ln(T)/t_\gamma} \quad (16)$$

where $t_\gamma$ is the time that bias $\gamma$ has been played.

In each turn, the agent first computes the UCB of all configurations with Eq. 13 and then selects the configuration with optimal UCB. Afterward, the agent evaluates the upper confidence bounds of all visualizations with Eq. 14 to select the optimal. Then, the agent will ask for user feedback on the recommended visualization: if it is positive, the agent will automatically take the configuration and attributes as positive; otherwise, it will further ask for user feedback on the configuration and attributes separately.

Intuitively, adding a bias term in the estimation of visualization reward can improve the accuracy of recommendation, because in the worst case we can assume it is the visualization reward and explore in a large action space. By designing appropriate reward function for the bias term, the bias term can serve as a correction term for cases that user likes the configuration and attributes but not the visualization. With the hierarchical structure of our agent, we further narrow down the large action space of the bias term. The agent quickly converges in configuration bandit with less item pool, so that it can have more exploration of attribute and bias terms with larger item pools.

### 4.2 Regret Analysis

The regret of Hier-SUCB comes from the exploration of preferred configuration, attribute pair and learning the bias term. The exploration of preferred configuration and attribute pair can be reduced to general combinatorial bandit problem. Learning bias term can be viewed as a general bandit problem with constraints. For more

---

**Algorithm 1:** Hier-SUCB

**1** Initialize $\theta_{C,t}, \theta_{A,t}, \gamma_t$ (Eq. 5);

**2** **for** $t = 1, 2, ...T$ **do**

**3**      **for** $a_{c,t} = 1, 2, ...n$ **do**

**4**          Compute UCB $U(c_t)$ (Eq. 13);

**5**      **end**

**6**      Select $c_t = \mathbf{argmax}(U(c_t))$;

**7**      **for** $a_{x,t} = 1, 2, ...m$ **do**

**8**          **for** $a_{y,t} = 1, 2, ...m$ **do**

**9**              Compute UCB $U(c_t, x_t, y_t)$ (Eq. 14);

**10**          **end**

**11**      **end**

**12**      Select $V_t = \mathbf{argmax}(U(c_t, x_t, y_t))$;

**13**      **if** $r_{V,t} == 1$ **then**

**14**          $r_{C,t} \leftarrow 1, r_{A,t} \leftarrow 1$;

**15**      **else**

**16**          ask for $r_{C,t}, r_{A,t}$;

**17**      **end**

**18**      Update $\theta_{C,t}, \theta_{A,t}, \gamma_t$ (Eq. 9);

**19**      Update $\rho_{c,t}, \rho_{a,t}, \rho_{\gamma,t}$ (Eq. 15, 16);

**20** **end**

---

detailed analysis of regret bound, we consider the regret of round $t$ under four cases when the user provides negative feedback to the visualization:

(1) Like configuration $c_t$ and attribute pair $\{x_t, y_t\}$
(2) Like attribute pair $\{x_t, y_t\}$ but not configuration $c_t$
(3) Like configuration $c_t$ but not attribute pair $\{x_t, y_t\}$
(4) Dislike configuration $c_t$ and attribute pair $\{x_t, y_t\}$

For case (1), we provide the regret bound by analyzing the bias term converges in certain rounds.

LEMMA 4.1. *The reward gap between optimal and sub-optimal bias $\gamma$ is bounded with the overall round $T$ and the time $t_\gamma$ that $\gamma$ has been played for.*

$$\Delta_\gamma \leq \sqrt{\frac{ln(T)}{t_\gamma}} \qquad (17)$$

PROOF. having a positive configuration and attributes while negative visualization implies:

$$U(c, a) \geq U(c^*, a^*) \qquad (18)$$

where $c^*, a^*$ refers to configuration and attributes of preferred visualization. Notably, $c, a$ may also receive positive feedback from user, but their combination is not preferred. In such case, $\Delta_c = \Delta_a = 0$, and we can bound the regret with bias:

$$\Delta_{bias} \leq \rho_{c,t} + \rho_{a,t} + \rho_{\gamma,t} - \rho_{c,t}^* - \rho_{a,t}^* - \rho_{t,\gamma}^* \qquad (19)$$

The round that $\gamma^*$ is updated given $c, a$ should be less than either $t_a^*$ or $t_c^*$, we define $t_{max}^* = max(t_a^*, t_c^*) \geq t_{min}^* = min(t_a^*, t_c^*) \geq t_\gamma^*$.

Using the definition of UCB, we can bound the gap of bias by

$$\Delta_{bias} \leq 3\sqrt{\frac{2ln(T)}{t_\gamma}} - 3\sqrt{\frac{2ln(T)}{t_{max}^*}} \leq 3\sqrt{\frac{2ln(T)}{t_\gamma}} \qquad (20)$$

$$t_\gamma \leq 18ln(T)\frac{1}{\Delta_{bias}^2} \qquad (21)$$

With 4.1, we can bound the regret bound of case (1) by:

$$Reg_1 = \mathbb{E}[t_\gamma]\Delta_{bias} \leq 18ln(T)/\Delta_{bias} = O(ln(T)) \qquad (22)$$

Notably, cases (2) and (4) are bounded by the rapid convergence of the confidence radius of configurations, thus, we consider when the agent recommends configuration. We derive the following lemma with $s_t^c$ representing the time that the configuration arm of action $a_t$ in round $t$ has been played.

LEMMA 4.2. *Following the proof in LinUCB [5], we can bound The gap between optimal and sub-optimal reward is bounded by the following equation with probability $1 - \delta/T$:*

$$|r_t^* - r_{t,a_t}| \leq \alpha\sqrt{-2log(\delta/2)/s_t^c} \qquad (23)$$

By summing Equation 23 with the expectation of round $T$, we derive the regret for case (2) and (4) as

$$Reg_{2,4} = O(\sqrt{Tln^3(m^2 Tln(T))}) \qquad (24)$$

For case (3), we first evaluate how many rounds the agent needs to recommend a positive configuration.

LEMMA 4.3. *With overall round $T$, the expected round for attribute exploration is*

$$T - \frac{k}{\Delta_c^2}ln(T) \qquad (25)$$

PROOF. The rounds to reach a positive configuration depend on the expectation of rounds that recommends a negative configuration. Thus by following the definition of UCB, we have:

$$\mathbb{E}[t] = k\frac{ln(T)}{\Delta_c^2} \qquad (26)$$

To calculate the regret bound of case (3), we apply the upper bound of round $t$ derived in Equation 23 and get:

$$Reg_3 = O(\sqrt{(T - ln(T))ln^3(m^2(T - ln(T))ln(T - ln(T)))}) \quad (27)$$

Therefore, we can get the overall regret by summing up the regret of each case:

THEOREM 4.4. *The regret of Hier-SUCB can be bounded as:*

$$Reg = Reg_1 + Reg_3 + Reg_{2,4} \qquad (28)$$

$$= O(\sqrt{Tln^3(m^2 Tln(T))}) \qquad (29)$$

Notably, we reduce the original semi-bandit by improving $O(nm^2)$ to $O(m^2)$ by adding a hierarchical structure and decompose the combinatorial problem to multi-arm bandits and contextual semi-bandits. Regular combinatorial contextual bandit will apply Eq. 24 to all the attributes and configurations, where the term $m^2$ would be $nm^2$ in this case. With a hierarchical structure, the regret of the

configuration is bounded by a contextual bandit. The regret of attribute pairs can be bounded with combinatorial contextual bandits as long as the configuration is preferred. For the case (4) where attributes and configuration are preferred but their combination is not, we model an independent bias as multi-armed bandits whose regret bound is $O(ln(T))$.

We also model the relation between the configuration and attribute with an extra bias as an individual bandit. This helps improve the final accuracy of the personalized visualization recommendation, which we demonstrate later in the experiments using real-world datasets.

## 5 EXPERIMENTS

In this section, we design experiments to investigate the following fundamental research questions:

**RQ1.** In a real-world dataset, is it a common case that a user likes a set of attributes and a specific configuration but dislikes the visualization resulting from the combination?

**RQ2.** Does the proposed bandit algorithm outperform the state-of-the-art offline method as well as other interactive methods that we adapted to our problem setting?

**RQ3.** Do the hierarchical structure and multi-armed bandit bias improve the performance?

### 5.1 Experimental Setup

In this section, we discuss our experimental setup. First, we introduce the baseline algorithms, then discuss the metrics used for evaluation.

*5.1.1 Baselines.* We compare our algorithm with two baseline algorithms that do not properly model the interrelation between the configuration and attributes in a visualization:

(1) **LinUCB** [5]: A non-combinatorial contextual bandit algorithm. This comparison demonstrates the Hier-SUCB ability to explore in a combinatorial setting.

(2) **C2UCB** [28]: A combinatorial contextual bandit algorithm. This comparison demonstrates the Hier-SUCB ability of more personalized and faster cold-start recommendation in PVisRec problem.

(3) **Neuro-PVR** [27]: A offline method trained for personalized visualization recommendation with neuro network. This comparison demonstrates the benefits of Hier-SUCB to provide personalized visualization recommendations with minimal samples.

*5.1.2 Metric.* We follow C2UCB [28] in the definition of cumulative regret and average reward. We get average reward of one round by computing the cumulative sum of the mean of the user feedback in each round. Similarly, cumulative regret is computed from the cumulative sum of the mean regret in each round. When comparing to the offline method Neuro-PVR [27], we compare the HR@K over iterations. Recall that HR@K (hit rate at $k$) is the fraction of the top $k$ recommended visualizations that are in the set of visualizations that are actually relevant to the user. We compare our method with the offline method Neuro-PVR [27] as well as with other bandit algorithms. For this comparison, we use HR@1 since the bandit algorithms mentioned are designed for recommending one

visualization. Furthermore, HR@1 is naturally the most important, since it indicates how likely the approaches are to recommend the desired visualization to the user directly without showing a sequence of visualizations.

### 5.2 Real-world Dataset

*5.2.1 Dataset Description.* For our experiments, we used the Plot.ly dataset curated by Qian et al [27]. There are improved dataset Plotly.plus [25] which provides better notation of user preference on visualizations, but for fair comparison with Neuro-PVR in [27], we keep the same dataset with Qian. The dataset contains information collected from the Plot.ly community, including the number of users, attributes, datasets, visualizations, and visualization configurations extracted from all the user-generated visualizations. The full corpus Plot.ly-full consists of 17469 users with 94419 datasets uploaded by these users. The corpus has a subset of 1000 datasets randomly, notated as Plot.ly-1k. We also take the attribute embedding generated by Qian from Plot.ly, representing the statistical features of attributes [27]. The embedding provided includes 10-dimensional vector, 30-dimensional vector and a 1004-dimensional vector. From the HR@k and NDCG@k metrics, we conclude that 10-dimensional vectors achieve the best performance and use them as the attribute embedding in our experiments.

*5.2.2 Preprocess.* Given the Plot.ly data, we further explore other features of the dataset. We derive a histogram as shown in Fig. 3, which describes the distribution of the amount of attributes in different user datasets. We observe a significant long-tail effect that most dataset has less than 100 attributes. Following the preprocess in [27], we filter out any dataset with more than 100 attributes. We list all the configurations in the dataset, which has a small armpool.

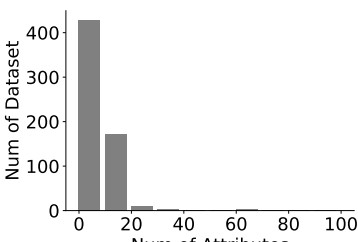

| All Possible Configurations | | | | |
|---|---|---|---|---|
| surface | scatter | scattergl | box | bar |
| mesh3d | scatter3d | contour | heatmap | histogram |

**Figure 3 & Table 1: Figure 3 shows the distribution of the number of attributes in different user datasets. Table 1 shows all possible configurations of the processed dataset. The item pool of configuration is relatively small**

*5.2.3 Simulator.* To evaluate the performance of our bandit algorithm, we need a simulator that resembles a real user and can react to the multiple-round recommendation from our agent. Similar to [23, 35], we build a simulator that gives Bernoulli feedback based on the setting of personalized visualization recommendation described previously. The user will provide three kinds of feedback: the feedback on attributes evaluating user preference for attribute pairs, the feedback on configurations evaluating user preference for the configurations, and the feedback on the entire visualization. The

agent receives different rewards separately to update the bandits of attributes and the configuration independently. We also add a noise term to the Bernoulli reward provided by the simulator, which gives users a 5% chance of providing the opposite response.

*5.2.4 User study of simulator.* To validate the performance of simulator in the real life, we perform a user study based on the visualization generated by algorithms. We first record the visualization generated by Hier-SUCB and C2UCB given the feedback of the simulator. Then we recruited 51 participants (university students aged from 18 to 26, 47 of them reported experience of creating visualizations) and recorded their preference on 10 samples randomly selected from those visualizations. As shown in Fig. 4 (a), we compare the percentage of user preference in round 20 and round 50 of the visualization generated in simulated experiments. From the human evaluation, we observe an increase in the population of participants that like either of the visualization, which implies the simulator can help bandit algorithms learn user preferences in the real-world setting.

## 5.3 Results

We implement some experiments on the real-world dataset and simulator introduced above. In the following sections, we will introduce how we implement the experiment and how the experiment result gives answers to the research question.

*5.3.1 A Study of Visualizations in the Dataset.* To address RQ1, we perform analysis on the Plot.ly-full dataset to observe whether it is a common case that user likes attributes and configuration but dislikes their visualization. For each user in the dataset, we find all the preferred attributes and configurations. Then we examine the ratio of their combinations in all possible combinations of attributes and configurations. As shown in Fig. 4 (b), the combination of preferred attributes and configurations only makes up for 4.1% of all combinations. We further examine the ratio of the preferred visualization in these combinations, and find only 22% of the combinations are liked by the users as visualizations. This observation indicates that when the agent learns from this dataset, it is highly likely that the

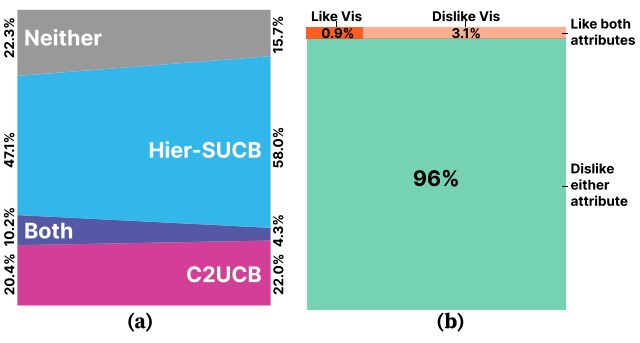

**Figure 4: (a) Our user study shows that from round 20 to 50, the percentage of users that like either visualization increases, indicating the simulator can help bandit algorithms learn user preference. (b) In the visualization dataset Plot.ly, even if a user prefers a set of attributes and visual configurations, they may not prefer their combination.**

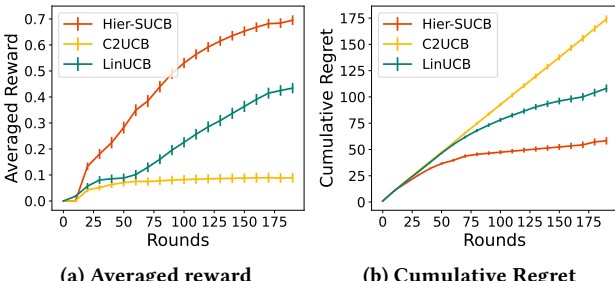

(a) Averaged reward    (b) Cumulative Regret

**Figure 5: Comparison of the hit rate using C2UCB, LinUCB, Hier-SUCB in the synthetic data over 100 iterations. Hier-SUCB outperforms other algorithms in 200 rounds.**

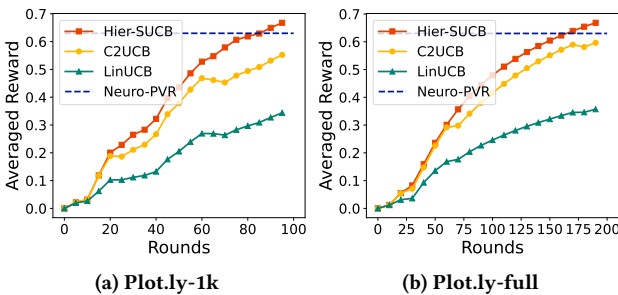

(a) Plot.ly-1k    (b) Plot.ly-full

**Figure 6: Comparison of the averaged reward (HR@1) using C2UCB, LinUCB, Hier-SUCB and Neuro-PVR (offline method). Hier-SUCB outperforms other bandit algorithms and exceeds the HR@1 of Neuro-PVR in round 80 and 160.**

configuration and attribute pair liked by the user will not build up a preferred visualization.

*5.3.2 Performance in Synthetic and Real-world Dataset.* In this section, we compare the performance of bandit algorithms and offline method called Neuro-PVR [27] to show the advantage of online methods. To answer RQ2, we compare the hit rate of bandit algorithms and the offline method in the synthetic and real-world dataset (Plot.ly-1k and Plot.ly-full). We start with building a synthetic setting to simulate the setting of personalized visualization recommendation. As shown in Fig. 5, the precision of Hier-SUCB and C2UCB are higher than others. Our method outperforms LinUCB and C2UCB with higher averaged reward. The experiment runs 200 rounds over 100 iterations.

We move forward to the real-world dataset Plot.ly. It runs 100 rounds for Plot.ly-1k and costs 0.01 second per round with Intel 13700K, and runs 200 rounds for Plot.ly-full. As shown in Fig. 6, in both datasets, C2UCB, Hier-SUCB converges and outperforms LinUCB. Also in the few shot setting, Hier-SUCB are higher than other two algorithms. In Plot.ly-1k, the hit rate of Hier-SUCB exceeds Neuro-PVR in 80 rounds, while in Plot.ly-full it exceeds Neuro-PVR in 160 rounds. The experiments validate that Hier-SUCB outperforms other bandit algorithms and the offline method.

We also conduct a case study of Hier-SUCB. As shown in Fig. 7, we compare visualization recommended by C2UCB and Hier-SUCB in round 1,10,20 and 50. All the visualizations are from the same subset of Plot.ly but evaluated by different users. It shows that the visualization recommended by Hier-SUCB are more likely to be preferred by users, indicating it is more personalized than C2UCB.

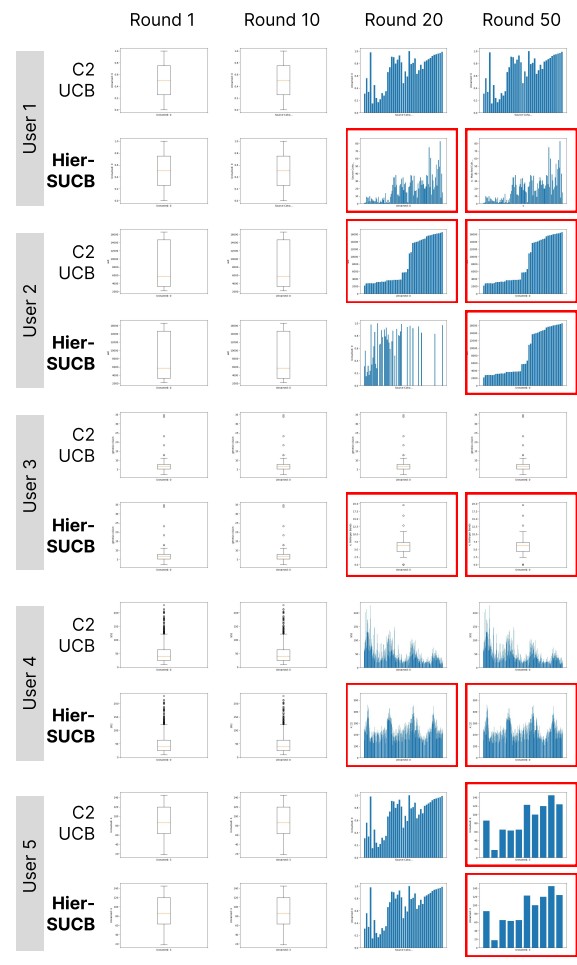

**Figure 7: A case study of the visualizations recommended by Hier-SUCB and C2UCB in round 1,10,20 and 50. The visualizations boxed by red rectangles are labeled as preferred by the corresponding user.**

*5.3.3 A Study of Hier-SUCB Variants.* In this section, we do experiments to answer how the hierarchical structure and the bias term in bandit design help the learning process (RQ3). We first compare LinUCB with Hier-SUCB in Fig. 6. In our experiment setting, LinUCB uses the combination of attributes and configuration as an arm. We observe LinUCB has slower convergence, lower averaged reward and higher cumulative regret over 200 rounds compared to Hier-SUCB. The observation validates our assumption in RQ3 that hierarchical structure helps the agent to learn faster.

To further confirm that the hierarchical structure contributes to the rapid convergence, we compare the average reward and cumulative regret of the original Hier-SUCB, a variation of Hier-SUCB without the hierarchical structure, and another variation that lacks the bias term in Plot.ly-full. In the variation without hierarchical structure, the agent no longer decides the configuration before deciding the visualization, so the visualizations are chosen from $O(nm^2)$ action space. In the variation without bias terms, the feedback provided by user is no longer used to update the bias term. In

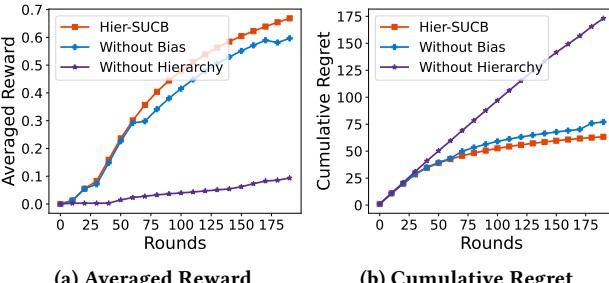

(a) Averaged Reward  (b) Cumulative Regret

**Figure 8: Comparison of the average reward and cumulative regret of Hier-SUCB and its variants Hier-SUCB without bias terms and Hier-SUCB without hierarchical structure. The performance drops when either component is missing.**

the experiments shown in Fig. 8b, we notice that the Hier-SUCB outperforms its variations with no configuration splitting in cumulative regret over 200 iterations. It is worth noting that Hier-SUCB without the bias term encounters hindrances around rounds 70 and 170. As a combinatorial contextual bandit, it is reasonable that Hier-SUCB without bias term is disturbed in the learning process by a biased estimated reward.

In this section, we do experiments in Plot.ly and its subset to show how the mapping from configuration to clustering helps the learning process. We compare the reward and regret curves of LinUCB, C2UCB and Hier-SUCB. In the experiments shown in Fig. 8, we notice that the Hier-SUCB outperforms its variation with no hierarchical structure and its variation with no bias term over 80 iterations in Plot.ly-1k. The experiments comparing Hier-SUCB with other bandit algorithms and offline method Neuro-PVR demonstrates the effectiveness of Hier-SUCB in PVisRec setting. The experiment comparing Hier-SUCB with its variants validates that hierarchical structure improves the exploration of visualization, and the bias term makes up for the gap between the real reward and estimated reward in combinatorial contextual bandits.

## 6 CONCLUSION

For a more personalized and sample-efficient recommendation of visualization, we formulate a novel combinatorial contextual semi-bandit with hierarchical structure and learnable bias term. To narrow the gap between the real reward and the estimated reward, we further model a learnable bias term which measures the relation between configuration and attributes. To apply the combinatorial semi-bandit to the PVisRec setting, we propose a hierarchical bandit structure that receives flexible feedback from users and provides reasonable feedback to the bias term. Combining the learnable bias term and hierarchical bandit structure, we proposed a new approach called Hier-SUCB. We perform a regret analysis on the approach and derive an improved overall regret bound of $O(\sqrt{Tln^3(m^2Tln(T))})$. Through a synthetic experiment and simulated experiments validated by human evaluation, we observe that Hier-SUCB exceeds offline methods in HR@1 in 80 rounds and outperforms traditional bandits with higher rewards and lower cumulative regrets. We also compare Hier-SUCB with its variants in a simulated experiment to demonstrate the effectiveness of the learnable bias term and hierarchical bandit structure.

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
