# OpenReview forum: "Interactive Visualization Recommendation with Hier-SUCB"
_ACM.org/TheWebConf/2025/Conference — WWW 2025 Oral_

### Official Review · Reviewer_hoFD · 2024-11-19

**Novelty:** 6
**Technical Quality:** 6

**Review:**

In this paper, the authors study an interesting and relatively rarely studied recommendation problem in the community of recommender systems, i.e., personalized visualization recommendation. The studied problem is well justified and the gap is clearly presented. Technically, the authors develop an interactive personalized visualization recommendation (PVisRec) system that is able to learn user preference from his or her historical interactions. Moreover, the authors provide some theoretical results (regret bound) for a proposed a contextual combinatorial semi-bandit (Hier-SUCB) in the PVisRec setting. Extensive empirical studies show the effectiveness of the Hier-SUCB. Overall, the paper is well presented.

**Questions:**

1 The authors use some simulator in the experiments. Is there some gap between the setting of the empirical study and that in a real-world application? The authors may include some discussions.

2 What will the time complexity or time cost/delay be in a real-world application using the proposed approach? The authors may include some discussions/analysis about the factors that will affect the time cost/delay, since that is important for real-time user-system interactions.

**Reviewer Confidence:**

3: The reviewer is confident but not certain that the evaluation is correct

**Scope:**

4: The work is relevant to the Web and to the track, and is of broad interest to the community

---

### Official Review · Reviewer_3dgN · 2024-12-01

**Novelty:** 5
**Technical Quality:** 4

**Review:**

## Overview
This paper introduces Hier-SUCB, a novel algorithm designed for interactive personalized visualization recommendation systems (PVisRec). It aims to address challenges of non-interactive systems, which rely on static user embeddings and cannot adapt dynamically to user feedback. The approach employs a hierarchical structure and introduces a learnable bias term in a contextual combinatorial semi-bandit framework, enabling faster adaptation and improved user personalization. Extensive experiments demonstrate that Hier-SUCB outperforms existing bandit algorithms and offline methods in cumulative reward, regret minimization, and hit rate across synthetic and real-world datasets.

## Strengths
- The hierarchical structure and learnable bias term enable effective modeling of the interaction between data attributes and visualization configurations, addressing a critical gap in existing combinatorial bandit approaches.
- The improved regret bounds and robust theoretical analysis enhance confidence in the applicability and effectiveness of the proposed algorithm.
- The paper rigorously benchmarks Hier-SUCB against several baselines, including state-of-the-art bandit algorithms and offline methods, across synthetic and real-world datasets.

## Weaknesses
- The computational efficiency of Hier-SUCB in scenarios with significantly larger datasets or complex visualization requirements is not fully explored.
- While the Plot.ly dataset provides a relevant benchmark, the findings may not generalize across other datasets or domains with fundamentally different visualization needs.
- The rationale behind the algorithm’s decision-making (e.g., why specific configurations are preferred) is not extensively discussed, which may hinder broader adoption.

**Questions:**

- How does Hier-SUCB scale computationally in real-world applications with significantly larger or more complex datasets?

- Could you provide more evidence or use cases showcasing the generalizability of Hier-SUCB across different domains or visualization tasks?

- How does the proposed bias term compare with alternative methods for modeling configuration-attribute relationships?

**Reviewer Confidence:**

1: The reviewer's evaluation is an educated guess

**Scope:**

3: The work is somewhat relevant to the Web and to the track, and is of narrow interest to a sub-community

---

### Official Review · Reviewer_WpCU · 2024-12-02

**Novelty:** 4
**Technical Quality:** 5

**Review:**

This work proposes an interactive visualisation recommendation system. It incorporates real-time user feedback and adapts a contextual combinatorial semi-bandit algorithm Hier-SUCB. To be more specific, it introduces a learnable bias term with hierarchical structure.

Pros:
- It clearly lists the challenges exist in most visualisation recommendation systems, and proposes a method to overcome these issues.
- It tries to verify both theoretically and via offline experiments on its effectiveness in terms of lower regret bound.
- The overall presentation is clear and easy to follow.

Cons:
- The algorithm used is mainly contextual combinatorial semi-bandit, which is not something new. Despite there are modifications to adapt it to visualisation recommendation, more technical novelty is expected.
- It seems the setting in the paper restricts to 2 categories 3 items, maybe it could be written or analysed in a more general/scalable form.
- In the offline experiment, it runs up to hundred rounds. I doubt how it could be used in real industry, as we could not expect a user to interact for so many rounds.
- For page4 line432, it would be better to move the term explanation to section 3 when it first appears.
- For Figure4(a), it is better to label which one refers to round 20.

**Questions:**

- Could this problem be easily addressed by using LLM? What could be the advantage/disadvantage comparing with the proposed method?
- Since the method claims its advantage when deal with larger action space, can you provide how it performs differently on dataset with more than 100 attributes?
- In real industry usage, we cannot expect the user to interact with the system for up to hundred of rounds, right?

**Reviewer Confidence:**

3: The reviewer is confident but not certain that the evaluation is correct

**Scope:**

4: The work is relevant to the Web and to the track, and is of broad interest to the community

---

### Official Review · Reviewer_7isq · 2024-12-02

**Novelty:** 6
**Technical Quality:** 5

**Review:**

The paper uses a hierarchical semi UCB framework for visualization recommendations. The improvement in the theoretical lower bound is proved using eqn. 17 and eqn. 25, and supported by simulations.
pros
The authors introduce a hierarchical structure that makes the recommendation interactive.
The authors carry out user preference experiments to validate their simulator.
cons
There are some grammar issues and typos. It doesn't really hurt the overall understanding of the paper, though.
The author should apply their HierSUCB with different number of configurations in experiments, now that they claimed that the lower regret bound is free of $n$.

**Questions:**

Results in Fig. 6 and Fig. 8 only show the average of 100 trials. One should include errors as well, say adding error bars, to call superiority over other algorithms. This is obvious through the apparent gap of the cumulative regret of the hierarchal models w.r.t. other sampling schemes that have yet reached the sublinear phase. I have gone through a few bandit learning papers, and it seems common in the field to not to have errors plotted. Anyway the authors may take it as a somewhat minor issue (that doesn't really count towards the overall rating).

**Reviewer Confidence:**

2: The reviewer is willing to defend the evaluation, but it is likely that the reviewer did not understand parts of the paper

**Scope:**

4: The work is relevant to the Web and to the track, and is of broad interest to the community